Effect of precipitation change on the photosynthetic performance of Phragmites australis under elevated temperature conditions

Teng Linhong 1
Liu Hanyu 1
Chu Xiaonan 1
Song Xiliang 2 sxl0424@126.com
Shi Lianhui 2 shilh@sdau.edu.cn
1 Dezhou University , Dezhou , China
2 Shandong Agricultural University , Taian , China
Spagnuolo Valeria
Electronic publication date: 2022 Mar 10
Publication date: 2022
Volume: 10
Electronic Location ID: e13087
Received 2021 Oct 26; Accepted 2022 Feb 17
Copyright: © 2022 Teng et al.
Copyright year: 2022
Copyright holder: Teng et al.
License: This is an open access article distributed under the terms of the Creative Commons Attribution License, which permits unrestricted use, distribution, reproduction and adaptation in any medium and for any purpose provided that it is properly attributed. For attribution, the original author(s), title, publication source (PeerJ) and either DOI or URL of the article must be cited.
License URL: https://creativecommons.org/licenses/by/4.0/

Keywords: Phragmites australis, Photosynthesis, Precipitation, Warming, Non-stomatal limitation, Protection mechanism

Funding: National Natural Science Foundation of China 32000402 and 32101365 Fund for Doctor Research of Dezhou University, China 2019xgrc26 This work was supported by the National Natural Science Foundation of China (Nos. 32000402 and 32101365) and the Fund for Doctor Research of Dezhou University, China (No. 2019xgrc26). The funders had no role in study design, data collection and analysis, decision to publish, or preparation of the manuscript.

==============================
Background

As a fundamental metabolism, leaf photosynthesis not only provides necessary energy for plant survival and growth but also plays an important role in global carbon fixation. However, photosynthesis is highly susceptible to environmental stresses and can be significantly influenced by future climate change.

Methods

In this study, we examined the photosynthetic responses of Phragmites australis (P. australis) to three precipitation treatments (control, decreased 30%, and increased 30%) under two thermal regimes (ambient temperature and +4 °C) in environment-controlled chambers.

Results

Our results showed that the net CO2 assimilation rate (Pn), maximal rate of Rubisco (Vcmax), maximal rate of ribulose-bisphosphate (RuBP) regeneration (Jmax) and chlorophyll (Chl) content were enhanced under increased precipitation condition, but were declined drastically under the condition of water deficit. The increased precipitation had no significant effect on malondialdehyde (MDA) content (p > 0.05), but water deficit drastically enhanced the MDA content by 10.1%. Meanwhile, a high temperature inhibited the positive effects of increased precipitation, aggravated the adverse effects of drought. The combination of high temperature and water deficit had more detrimental effect on P. australis than a single factor. Moreover, non-stomatal limitation caused by precipitation change played a major role in determining carbon assimilation rate. Under ambient temperature, Chl content had close relationship with Pn (R2 = 0.86, p < 0.01). Under high temperature, Pn was ralated to MDA content (R2 = 0.81, p < 0.01). High temperature disrupted the balance between Vcmax and Jmax (the ratio of Jmax to Vcmax decreased from 1.88 to 1.12) which resulted in a negative effect on the photosynthesis of P. australis. Furthermore, by the analysis of Chl fluorescence, we found that the xanthophyll cycle-mediated thermal dissipation played a major role in PSII photoprotection, resulting in no significant change on actual PSII quantum yield (ΦPSII) under both changing precipitation and high temperature conditions.

Conclusions

Our results highlight the significant role of precipitation change in regulating the photosynthetic performance of P. australis under elevated temperature conditions, which may exacerbate the drought-induced primary productivity reduction of P. australis under future climate scenarios.

Introduction

Global warming mainly caused by high levels of greenhouse gas emission is predicted to increase the air temperature by 1.1–6.4 °C in the next hundred years (Crowther et al., 2016). At the same time, extreme precipitation events like drought and waterlogging will occur more universally than ever (IPCC, 2019). The changing global climate will not only aggravate the frequency and intensity of environmental stresses but also pose serious threat n agriculture production (Hossain et al., 2021; Vaughan et al., 2018; Xin & Tao, 2021), ecosystem stability (Kanojia & Dijkwel, 2018; White et al., 2021) and terrestrial C and N cycling (Crowther et al., 2016; Li et al., 2021b). Among the environmental factors, ambient temperature and soil water content are two major abiotic factors in the limitation of plant distribution and productivity (Küsters et al., 2021; Yan, Zhong & Shangguan, 2020; Kumari et al., 2021). Their change will directly and/or indirectly influence plant physiological processes, such as resource allocation (Farfan-Vignolo & Asard, 2012; Forbesa et al., 2020), net photosynthetic rate (Shao et al., 2021; Yamori, Hikosaka & Way, 2014), carboxylation efficiency (Liu et al., 2022), photochemical efficiency of photosystem II (PSII) (Aragón-Gastélum et al., 2020; Song et al., 2016a) and water use efficiency (Liu et al., 2019), which then impact the global carbon cycling. Among all the plant physiological processes, photosynthesis plays an important role in substance metabolism (Ort et al., 2015; Zhu et al., 2020). Thus, the understand of how plant photosynthesis responses to the concurrent warming and precipitation change is necessary for plants better facing future climate change.

The high limitation on the plant carbon assimilation capacity under soil water deficient conditions has been a major reason for plant growth and crop productivity reduction (Hussain et al., 2021; Nolf et al., 2015). It is widely accepted that there are two ways in which water stress affects the photosynthesis of plants: one is the stomatal limitations, such as closing the stoma and lowering the stomatal conductance (Daryanto, Wang & Jacinthe, 2017; Talbi et al., 2020); the other is non-stomatal limitations, such as photosynthetic phosphorylation (Du et al., 2021), regeneration of ribulose-1,5-bisphosphate (RuBP) (Song et al., 2016a), activation of Rubisco and the synthesis of ATP (Ashraf & Harris, 2013; Hu et al., 2020). The stomatal limitation is generally considered as the main factor responsible for the reduction of photosynthesis under drought stress environment (Liu et al., 2005; Song et al., 2020). However, long term of drought stress may lead to the reduction of chlorophyll content (Bijanzadeh, Barati & Egan, 2022), the content of Rubisco (Gadzinowska et al., 2021), the maximum Rubisco carboxylation rate and potential maximum rate of electron transport for RuBP regeneration (Song et al., 2016a), resulting in the decline of the plants’ photosynthetic rate (He et al., 2021; Wang et al., 2019).

The photosynthesis of plants is regarded as the most sensitive process to high temperature stress (Xalxo et al., 2020). High temperature lasting for only a few minutes to several hours will drastically damage the structure and function of photosynthetic apparatus such as thylakoid lamella and stroma, decrease the production of ATP, inhibiting a series of enzyme activities, affect the transport of photosynthetic electrons and reduce the photosynthetic rate finally (Hu et al., 2020). Heat stress can also cause photosynthesis decline through enhancing the generation of reactive oxygen species (ROS) (Hao et al., 2019), destroying the function of PSII (Jahan et al., 2021; Janka et al., 2015) suppressing the synthesis of chloroplast (Song, Wang & Lv, 2016), and inhibiting the activity of ribulose1,5-bisphosphate carboxylase/oxygenase (Rubisco) (Perdomo et al., 2017). In tomato plants, heat stress (40 °C) significantly decreased photosynthetic pigment concentrations and inhibited Rubisco accumulation resulting in a reduction of photosynthetic efficiency (Parrotta et al., 2020). Based on a 3-year study, Zhong et al. (2014) also reported that an air temperature elevation of 1.5 °C could decreased the net photosynthetic rate of Phragmites australis by 28%. In contrast, a recent study showed that increase of 4 °C significantly increased the net photosynthesis rate, transpiration rate, leaf temperature and chlorophyll content in leaves of lettuce by 114.9%, 65.5%, 7.1% and 9.8%, respectively (Ouyang et al., 2020). Although an emerging pool of knowledge shows that plant photosynthesis was noticeable affected by heat stress, the mechanism of the photoinhibition caused by high temperature is still need further research.

Coastal wetlands account for 0.22–0.34% of global land surface (Fennessy, 2014) and act as “blue carbon” resources due to the relatively high net primary productivity and low organic matter decomposition rate (Drake et al., 2015; Zhong et al., 2016). It is estimated that 13–17.2 Pg of carbon were stored in coastal wetlands (Hiraishi et al., 2014). However, coastal wetlands are also potential source of global greenhouse gases (Hsieh et al., 2020). The climate change increased the release rate of carbon in the CO2 and CH4 through organic matter decomposition and decreased the amount of carbon stored in coastal wetlands. It is found that a 1.5 °C temperature enhancement could result in the gas emissions released form wetlands increase by 37.5% (Liu et al., 2020). As plant photosynthesis is the major way of carbon fixation in coastal wetlands, keep the photosynthesis at a high rate under climate change conditions is essential for global carbon cycling. Phragmites australis (P. australis) belonging to the Poaceae family, is the main constructive and dominant plants in coastal wetlands of China and plays an important role in maintaining the ecosystem function (Guan et al., 2017). Their spatial distribution is mainly limited by air temperature change and soil water deficit. The research on P. australis’s photosynthetic characteristics in response to rising temperature and changing precipitation pattern can provide a theoretical basis for dealing with climate change in coastal wetlands. This main aims of the work were to investigate the photosynthetic responses of P. australis to precipitation change under elevated temperature conditions. Specifically, three key research questions were addressed in the paper: (1) Are there any negative or positive influences of temperature and precipitation change on photosynthetic performance of P. australis? (2) What are the physiological mechanisms of precipitation change and high temperature affecting the carbon assimilation of P. australis? (3) What are the protection mechanisms of P. australis to avoid damage caused by environmental stress?

Materials and Methods

Plant culture and experimental design

The experiment was carried out at the Dezhou University, Shandong Province, China. The seeds of P. australis and soils were obtained from the costal wetland in Kenli, Dongying, China. The soil sample site has a northern subtropical marine monsoon climate. The annual average temperature and precipitation which obtained from the Kenli Meteorological Station of the China Meteorological Administration (37°35′N, 118°33′E; elevation 85 m) in the past 10 years (2010–2019) were 12 °C and 552 mm, respectively. About 70–74% of the annual precipitation is concentrated from July to September.

Before sowing in plastic pots, the seeds of P. australis were sterilized by potassium permanganate solution (0.7%) for 8 min and washed with deionized water for three times. Each plastic pot (18 cm in height and 20 cm in diameter) was filled with 5.0 kg of dry soil and planted with 10 plants. The experimental soil was paddy fluvo-aquic soil, and the basic physical and chemical properties of the soil were as follows: soil pH 7.91, organic matter 9.42 g·kg−1, total nitrogen 0.77 g·kg−1, available phosphorus 5.92 g·kg−1, and available potassium 168.72 g·kg−1.

After the third leaf emerged, the seedlings were thinned to three plants per pot. There were three precipitation treatments and two temperature treatments were selected for experiment. The precipitation treatments were set as: average monthly precipitation (July to September) over 10 years (W0); W0 increased by 30% (W+30); W0 decreased by 30% (W−30). The temperature treatments were set as 26.3/21.6 °C (T0) and 30.3/25.6 °C (T4). The treatments were set based on the monthly average temperature and rainfall during P. australis’s major growth stage (July to September) in the past 10 years (2010–2019). Each treatment and corresponding experiments were established in triplicates. Totally, 18 pots with healthy plants (three plants per pot) were randomly selected and placed into two environmental control chambers (RGD-500D3). The size of environmental control chamber was 750 × 660 × 2,050 mm (length × width × height). Growing conditions in the environmental control chamber were maintained as follows: 390 ppm CO2 concentration, 1,000 µmol photons·m−2·s−1 photosynthetic photon flux density, and 14 h photoperiod per day. All the parameter measurements were conducted after 92 days of plant growth.

Measurements

Leaf gas exchanges

Three plants from each treatment were randomly chosen from different pots for measurement. Gas exchange parameters were measured on the healthy and fully expanded leaves of P. australis with an open gas exchange system (CIRAS-3, PP-system, Hitchin, UK). Illumination was supplied to the leaves from a red-blue LED light source. The leaf chamber temperature, CO2 concentration and photosynthetic photon flux density (PPFD) were controlled at 25 °C, 390 ppm and 900 μmol·m−2·s−1, respectively.

A/Ci curve

The measurement of A/Ci curves was performed on the same leaves used for gas exchange parameter measurements. A/Ci curve was measured under a light saturation level of 900 μmol·m−2·s−1 PPFD, and estimated using the CO2 response curve of photosynthesis. The CO2 gradients for A/Ci curves included 390, 200, 100, 50, 390, 600, 800, 1,000 μmol·mol−1 levels stepwise. The analysis of A/Ci curve was conducted with using the plant ecophys R package, which based on the model of Farquhar, Von & Berry (1980).

Chlorophyll fluorescence measurements

Three areas of interest at different position of leaf were selected to calculate the fluorescence parameters. Based on the method described by Song et al. (2016b), the actual PSII quantum yield (ΦPSII), quantum yield of regulated energy dissipation of PSII (ΦNPQ), and quantum yield of nonregulated energy dissipation of PSII (ΦNO) were measured using an imaging-PAM fluorometer (Walz, Effeltrich, Germany). The fluorescence parameters were calculated using fellow equations described by Lazár (2015):

ΦPSII=(Fm′−Fs)/Fm′=ΔF/Fm′ΦNPQ=1−ΦPSII−1/[NPQ+1+qL(Fm/F0−1)]ΦNO=1/[(NPQ+1+q(L))(Fm/F0–1)]

where Fm is the maximum fluorescence in the dark-adapted state, F0 is the minimum Chl fluorescence yield, Fm′ is the maximum fluorescence yield in the light-adapted state, Fs is the Chl fluorescence during actinic illumination, qL is the fraction of open PSII centers, NPQ is the non-photochemical quenching.

Chlorophyll content

The chlorophyll content was measured according to the method described by Hiscox & Israelstam (1979). Briefly, 0.25 g fresh leaf samples were mashed in 80% acetone (v/v) in a 4 °C refrigerator overnight. After filtered through two-layer nylon net, the extract was then centrifuged at 15,000 g for 5 min to obtain the supernatant. After determining the absorbance of the supernatant at wavelengths of 663 and 646 nm, the contents of chlorophyll a and b were calculated according to the equations of Lichtenthaler & Buschmann (2001):

Chlorophyll a=12.25 A663−2.79 A647Chlorophyll b=21.50 A647−5.10 A663

Malondialdehyde (MDA) content

The MDA content was measured according to the thiobarbituric acid (TBA) chromogenic method described by Song, Wang & Lv (2016). Briefly, 1.0 g fresh leaf samples were homogenized with 0.1% trichloroacetic acid (TCA, 2.0 mL, pH 7.0) for 2 h an 15,000 g for 10 min. Then, 0.5 mL of supernatant was added to 1.5 mL of TBA. After the mixture was incubated in a shaking water bath at 90 °C for 20 min, the reaction was rapidly stopped by ice-water bath. These samples were centrifuged at 10,000 g for 5 min to obtain the supernatant. The absorbance of the supernatant was detected at 532, 450, and 600 nm. The amount of MDA was calculated with the following equation:

MDA=6.45×(𝐴532−𝐴600)−0.56×𝐴450

Statistical analysis

All statistical analyses were performed using SPSS 21.0 (SPSS Institute, Inc., Cary, NC, USA). Effects of warming and precipitation change were analyzed using one-way analysis of variance with a Duncan’s multiple range test at a 5% probability level. The linear curve fitting and graphing were performed using Origin 2021 software (Origin Lab, Northampton, MA, USA).

Results

Chlorophyll content

Under both two temperature conditions (T0 and T4), the Chl content of P. australis was significantly affected by precipitation change. It can be seen from Table 1 that, at the condition of T0, W+30 caused the increase of Chl a, Chl b and Chl a+b content by 25.6%, 33.8% and 31.1%, respectively, with the Chl a/b ratio decreased by 6.6%. At the same temperature, a decreasing precipitation (W−30) led to the decline in Chl b and Chl a+b content (10.1% and 6.2%, respectively) and the increase in Chl a/b ratio (12.2%). At a higher temperature (T4), the adjustment of precipitation resulted in similar variations in the contents and ratios of Chl contents. Moreover, under different precipitation conditions (W+30, W0 and W−30), the Chl a, Chl b and Chl a+b content at the higher temperature (T4) decreased by 3.5–13.0%, 18.7–32.0% and 12.9–24.2%, respectively, with the Chl a/b ratio increasing by 7.6–25.3%.

Table 1 Effects of warming and precipitation changes on chlorophyll content in leaves of Phragmites australis.

Treatment	Chl a (mg/g)	Chl b (mg/g)	Chl a/b (%)	Chl a + Chl b (mg/g)	
T0	W+30	1.08 ± 0.05 a	1.86 ± 0.08 a	58.2 ± 0.7 b	2.95 ± 0.13 a	
W0	0.86 ± 0.04 b	1.39 ± 0.02 b	62.3 ± 1.7 b	2.25 ± 0.06 b	
W−30	0.86 ± 0.01 b	1.25 ± 0.09 c	69.9 ± 4.7 a	2.11 ± 0.10 b	
T4	W+30	0.94 ± 0.01 a	1.51 ± 0.05 a	62.6 ± 2.4 c	2.45 ± 0.04 a	
W0	0.83 ± 0.02 b	1.13 ± 0.10 b	74.0 ± 4.4 b	1.96 ± 0.12 b	
W−30	0.75 ± 0.01 c	0.85 ± 0.04 c	87.6 ± 3.8 a	1.60 ± 0.05 c	
Note:

Different lowercases indicate significant difference between different precipitation treatments within the same temperature treatment compared with control (p < 0.05).

MDA content

Malondialdehyde (MDA) as a product of lipid peroxidation can be used as a marker for oxidative stress under environmental stress conditions. The higher MDA content indicates the stronger cell membrane lipid peroxidation. It can be seen from Fig. 1 that, under both T0 and T4 conditions, W+30 had no significant effect on MDA content (p > 0.05). But W−30 led to the significant increase of MDA by 10.1% under T0 condition and by 9.5% under T4 condition. At the same time, high temperature also enhanced MDA content. As shown in Fig. 1, under different precipitation conditions (W+30, W0 and W−30), the MDA content in the T4 treatment groups increased by 5.2%, 6.3% and 5.7%, respectively, compared with the T0 treatment groups.

Figure 1 Effects of warming and precipitation changes on Malondialdehyde content in leaves of Phragmites australis.

Vertical bars represent ±SD of the mean (n = 3), and different letters on the SD bars indicate significant differences among the all treatments (p < 0.05).

Photosynthetic parameters

From Table 2, it was found that at the condition of T0, compared to W0, net CO2 assimilation rate (Pn) in W+30 treatment increased by 32.8% and in W−30 treatment reduced by 18.9%, respectively. The other gas exchange parameters such as stomatal conductance (Gs), intercellular CO2 concentration (Ci), transpiration rate (Tr) and water use efficiency (WUE) were not significantly affected by W+30 or W−30. At the condition of T4, the photosynthetic parameters between W+30 and W0 showed no remarkable difference, while W−30 significantly reduced the values of WUE, Pn, Gs, Ci and Tr by 25.2%, 52%, 14.1% and 33.0%, respectively. Under all precipitation conditions, high temperature negatively affected photosynthesis of P. australis and reduced Pn by 6.6∼17.4%.

Table 2 Effects precipitation change on photosynthetic parameters in leaves of Phragmites australis under ambient temperature (T0) and high temperature (T4) conditions.

Treatment	Photosynthetic parameters	
	Pn (μmol CO2·m−2·s−1)	Gs (μmol·mol−1)	Ci (mol H2O·m2·s−1)	Tr (mmol·m−1·s−1)	WUE (μmol CO2·mmol H2O)	Vcmax (μmol·m2·s−1)	Jmax (μmol·m2·s−1)	
T0	W+30	11.5 ± 1.0 a	0.18 ± 0.03 a	271 ± 12 a	3.5 ± 0.3 a	3.3 ± 0.3 a	52.0 ± 8.1 a	121.0 ± 23.1 a	
W0	8.6 ± 0.4 b	0.14 ± 0.08 a	224 ± 69 a	3.7 ± 1.8 a	2.7 ± 1.3 a	41.5 ± 1.8 b	75.5 ± 11.7 ab	
W−30	7.0 ± 0.7 c	0.15 ± 0.01 a	303 ± 2 a	3.4 ± 0.3 a	2.1 ± 0.4 a	27.1 ± 5.4 c	69.3 ± 18.6 b	
T4	W+30	9.5 ± 0.1 a	0.14 ± 0.01 a	267 ± 2 a	3.5 ± 0.1 a	2.7 ± 0.1 a	58.1 ± 6.5a	110.6 ± 6.0 a	
W0	8.1 ± 0.7 a	0.17 ± 0.03 a	298 ± 15 ab	3.0 ± 0.3 a	2.7 ± 0.01 a	35.4 ± 4.6 b	79.0 ± 4.1 b	
W−30	6.0 ± 1.2 b	0.08 ± 0.01 b	256 ± 24 b	2.0 ± 0.6 b	3.2 ± 1.3 a	25.7 ± 3.8 b	56.5 ± 10.0 b	
Note:

Different lowercases indicate significant difference between different precipitation treatments within the same temperature treatment compared with control (p < 0.05).

The change of Pn as a function of increased Ci in the chloroplast can be used to reflect the biochemical limitations of photosynthesis under high temperature and changing precipitation conditions. As shown in Table 2, at the condition of T0, W+30 enhanced Vcmax and Jmax by 25.3% and 60.3%, while W−30 caused the reduction by 34.6% and 8.2%, respectively. At the condition of T4, W+30 resulted in a significant increase of Vcmax and Jmax by 63.8% and 27.3%, while W−30 caused the reduction by 27.4% and 28.4%, respectively. Under W0 and W−30 conditions, T4 significantly reduced Vcmax by 14.5% and 5.1%, while increased Jmax by 4.7% and 18.4%, respectively. At the condition of W+30, Vcmax increased by 11.7% and Jmax decreased by 16.9% in the T4 treatment group.

Chlorophyll fluorescence parameter

The effect of water treatments on ΦPSII, ΦNPQ and ΦNO under two temperature conditions were shown in Fig. 2. Under T0 condition, precipitation change (W+30, W−30) had no significant effect on ΦPSII (p > 0.05), but drastically increased ΦNPQ by 14.9% and 32.3% and reduced ΦNO by 13.3% and 22.7%, respectively. Under T4 condition, ΦPSII in the W+30 and W−30 treatment groups increased by 8.6% and 6.8%, ΦNO increased by 30.3% and 21.3%, while ΦNPQ decreased by 25.4% and 18.9%, respectively. Under different precipitation treatments (W+30, W0 and W−30), compared to T0, the change of ΦPSII caused by T4 was 29.6%, −4.1% and 9.3%, the change of ΦNPQ caused by T4 was −10.6%, 36.8% and −16.0%, and the change of ΦNO caused by T4 was 5.8%, −29.7% and 11.6%, respectively.

Figure 2 Effects of warming and precipitation changes on ΦPSII (A), ΦNPQ (B) and ΦNO (C) in leaves of Phragmites australis.

The horizontal line represents the median value and the open rectangle represents the mean value (n = 9). *p<=0.05, **p<=0.01, ***p<=0.001.

Discussion

High temperature and precipitation change as two major abiotic stresses always occur simultaneously, which threaten the sustainability of future crop production and biodiversity (Alam et al., 2021; Hosseini Sanehkoori, Pirdashti & Bakhshandeh, 2021; Küsters et al., 2021; Zhang et al., 2018). In the present study, we found that the positive effects of increased precipitation and the adverse effects of decreased precipitation on chlorophyll content, CO2 assimilation rate, lipid peroxidation (as indicated by MDA) and the energy partitioning of PSII were significant. Meanwhile, high temperature inhibited the positive effects of increased precipitation and aggravated the adverse effects of decreased precipitation. Similarly, in the studies on Leymus chinensis (Xu & Zhou, 2011), Stipa bungeana (Song et al., 2016b), Ziziphus jujube (Jiang et al., 2020), and Robinia pseudoacacia (Yan, Zhong & Shangguan, 2020), the high temperature combined with severe drought exacerbated the adverse effects on plant growth and photosynthesis.

Plants exposed to environmental stresses, such as drought, extreme temperatures or their combinations, have to face several metabolic imbalances leading to oxidative damage due to ROS accumulation, resulting in detrimental secondary effects on plant organelles (Raja et al., 2020; Vurukonda et al., 2016). ROS buildup in plants can damage cell functions by causing oxidative damage, resulting in DNA nicking, amino acids and photosynthetic pigments biosynthesis inhibition, and even cell death (Nath et al., 2016; Raja et al., 2017). MDA content, a result of ROS mediated lipid peroxidation, is used as biomarker of membrane damage caused by various abiotic stresses (Morales & Munné-Bosch, 2019). In the present study, increased precipitation showed no significant effect on MDA content in leaves of P. australis, while the decreased precipitation and elevated temperature remarkable increased the MDA content. The results suggest precipitation decreased by 30% and temperature elevated by 4 °C accelerates MDA formation, resulting in serious lipid peroxidation (Morales & Munné-Bosch, 2019). Similar results were found in studies on Solanum lycopersicum (Raja et al., 2020), maize (Naz et al., 2021), and Echinacea purpurea (Hosseinpour et al., 2020). The increase in the MDA content indicates that water deficit and high temperature destroy the antioxidant defense system, generate lipid peroxidation, and cause oxidative burst and excess oxidative damage to the cell membrane in P. australis plants. The increase in lipid peroxidation is widely reported to cause oxidative damage to chloroplast organs (Sohag et al., 2020) and leads to chlorophyll degradation (Bagheri, Gholami & Baninasab, 2019). The noticeable reduction of Chl a and Chl b in the W−30 and T4 treatment supports the finding that water deficit and high temperature trigger oxidative damage to the expression of chlorophyll a-b binding protein gene (Sun et al., 2022) and the synthesis of chlorophylls (Gujjar et al., 2020), which inevitably leads to a decrease in leaf photosynthetic efficiency (Wang et al., 2019) and plant productivity (Song, Jin & He, 2019).

The response of photosynthetic capacity to the variation of soil water depends on the threshold of soil water condition. Lamptey et al. (2020) and Snider et al. (2014) proved that the photosynthetic activity will be enhanced under moderate soil water condition but be lowered under excess water or severe water deficit conditions. In the present study, increased precipitation (W+30) did not exceed the threshold of soil moisture and significantly increased the value of Pn. This suggests that the precipitation increased by 30% is a moderate soil water condition for the potential photosynthetic capacity of P. australis. The reduction of Pn at the W−30 condition demonstrated that the severe drought stress can drastically inhibit the photosynthesis of P. australis. At the same time, previous studies also showed that the photosynthesis and plant growth will be limited by higher temperature above the optimum point (Rodriguez et al., 2015). In our study, the reduction of Pn under T4 condition indicated that the temperature 4 °C higher than the ambient temperature (26.3/21.6 °C) has exceeded the optimum point and is adversely to the photosynthesis of P. australis. However, the threshold of soil water condition and the optimum temperature point for the photosynthesis of P. australis are still unclear and need further investigation. It is widely accepted that the decline in Pn, Ci, Tr and WUE could be attributed to decreased Gs under drought and heat stress conditions (Carvalho et al., 2019; Li et al., 2021a; Olorunwa, Shi & Barickman, 2021). In this study, under ambient temperature (T0) condition, the Gs, Ci, Tr and WUE showed no remarkable differences in different precipitation treatments, indicating the soil water deficit is not the limiting factor in stomatal openness, water consumption (transpiration) and utilization for P. australis plants. On the other hand, with the increasing of temperature (T4), precipitation decreased by 30% caused a remarkable reduction of Gs, Ci and Tr, suggesting that higher temperature exacerbates the detrimental effect of water shortage, which is in accordance with the studies on Xanthoceras sorbifolium Bunge (Du et al., 2021), Solanum lycopersicum (Raja et al., 2020), and Stipa bungeana (Song et al., 2016b). Furthermore, drought and heat stress also cause damage to the photosynthetic apparatus as confirmed by reduced Vcmax and Jmax, as the decline in these two parameters are ascribed to a reduced number of active Rubisco molecules and a decrease of photosynthetic energy during the process of CO2 assimilation (Olorunwa, Shi & Barickman, 2021; Zhuang et al., 2020).

The mechanisms of precipitation change affecting the carbon assimilation can be studied by stomatal limitation and non-stomatal limitation. Song et al. (2020) indicated that the reduction in photosynthesis of a water-stressed maize was mainly caused by stomatal limitation, whereas Li et al. (2020) reported that stomatal limitation did not play a major role in the change of photosynthesis of transgenic tobacco plants. The different results may be attributed to various responses from species, stress lasting time and stress intensity (Mitchell et al., 2008; Song et al., 2020). In our experiment, to figure out which is the main factor in limiting the photosynthesis, linear regression analysis was performed to illustrate the relationship of Pn with Gs, Vcmax, Jmax, Chl a+b content, Chl a/b ratio and MDA content under T0 and T4 conditions, respectively (Fig. 3). From the linear regression analyses, it was found there is no significant relationship between Pn and Gs (p > 0.05). But Pn had a significantly positive linear correlation with Vcmax, Jmax and Chl a+b content, as well as a significantly negative linear correlation with Chl a/b ratio and MDA content. The results indicate that non-stomatal limitation caused by precipitation change plays a major role in determining the carbon assimilation rate. Similar result can be found in the research by Xu & Zhou (2011), Song et al. (2016a), and Li et al. (2020). At the condition of T0, Chl a+b content had the closest relationship with Pn (R2 = 0.86, Fig. 3D) compared with other non-stomatal limitation factors. This suggests that the effect of increased precipitation on Chl content plays a major role in determining the carbon assimilation under ambient temperature condition. At the condition of T4, MDA content had the closest relationship with Pn (R2 = 0.81, Fig. 3F) compared with other non-stomatal limitation factors. This suggests that the effect of increased precipitation on lipid peroxidation plays a major role in determining the carbon assimilation under high temperature condition.

Figure 3 Relationship between Pn and (A) Gs, (B) Vcmax, (C) Jmax, (D) Chl a+b, (E) Chl a/b and (F) MDA content under ambient temperature (T0) and high temperature (T4) conditions.

In the present study, we found that high temperature induced the stomatal opening (increase in Gs, Table 2), but resulted in a decrease in carbon assimilation (decrease in Pn, Table 2), which is consistent with the research on Leymus chinensis by Xu & Zhou (2006). The response mechanism of plant photosynthesis to temperature can be studied by the balance between Vcmax and Jmax (And & Sharkey, 1982; Song et al., 2016b). Wullschleger (1993) investigated 109 different species and concluded that there was a strong correlation between Vcmax and Jmax, which means there was a fixed balance relationship between RuBp carboxylation and regeneration in spite of the species or growth conditions. In our study, Vcmax and Jmax showed a significant linear relationship under ambient temperature (T0) condition, with the ratio of Jmax to Vcmax being 1.88 (p < 0.05, Fig. 4A). However, with the increasing of temperature (T4), even though there was still an obvious linear relationship between Vcmax and Jmax (p < 0.05, Fig. 4B), the ratio of Jmax to Vcmax decreased to 1.12. These results indicate that high temperature disrupted the balance between Vcmax and Jmax, resulting in a negative effect on the photosynthesis of P. australis. Similar results were also supported by the study of Huang et al. (2021).

Figure 4 Relationship between the maximum rate of RuBP carboxylation (Vcmax) and RuBP regeneration capacity (Jmax) in leaves of Phragmites australis under warming and precipitation change conditions.

Chlorophyll fluorescence can be used to detect the real photosynthetic behavior of the whole plant under stress quickly (Bhagooli et al., 2021). Based on this, it is possible to evaluate both the function of photosynthetic apparatus and the effects of environmental stress on plants. Environmental stress mainly damages the photosynthetic apparatus of PSII, and PSII will adjust the rate of electron transport and photochemical efficiency in response to the weakened ability of CO2 assimilation (Aragón-Gastélum et al., 2020; Hasanuzzaman et al., 2013). The damage caused by excess light energy to the system will be mitigated by heat dissipation. Water deficiency and heat stress will cause the inactivation or damage of leaf’s PSII reaction center (He et al., 2021; Mathur, Agrawal & Jajoo, 2014). This will lead to the damage of the photosynthetic apparatus and bring about the photoinhibition, which is consistent with the studies by Farfan-Vignolo & Asard (2012) and Yan et al. (2018). In our present research, precipitation change and high temperature had a significant effect on the photosynthesis of P. australis. However, how P. australis resists those environmental stresses to protect itself is still unknown. To solve this problem, three fluorescence parameters (ΦPSII, ΦNPQ and ΦNO) based on Lake model were used to detect the partitioning of absorbed light energy and to explore the protective mechanism in PSII reaction center (Kramer et al., 2004; Li et al., 2019b). Among the three fluorescence parameters, ΦPSII (absorbed light energy utilized by PSII photochemistry) reflects the linear electron transport indirectly, ΦNPQ (thermally dissipated via ΔpH and xanthophyll-dependent energy quenching) represents the yield of dissipation by downregulation, and ΦNO (thermally dissipated via ΔpH and xanthophyll-dependent energy quenching) reflects the yield of other non-photochemical losses (García-Sánchez et al., 2012; Nabi et al., 2021). In Figure 2, it was found that precipitation change and high temperature had no significant effect on the value of ΦPSII, suggesting that heat dissipation of the excess light energy was dissipated to the extracelular as a form of heat to protect the photosynthetic apparatus from damage caused by photoinhibition (Li et al., 2019a; Song et al., 2016b). Moreover, Figure 5 showed that there was a strong relationship (p < 0.01) between ΦPSII and ΦNPQ, and the correlation between ΦPSII and ΦNO were not evident (p > 0.05). This suggests that the xanthophyll cycle-mediated thermal dissipation plays a major role in PSII photoprotection under changing precipitation and high temperature conditions, while the non-regulated quenching mechanism may play a less important role (Demmig-Adams & Adams, 2018; Stael et al., 2015). The results are opposite with the findings on plant responses to heat stress, water deficit and cold stress by other scholars (Dias et al., 2018; Osório et al., 2011; Savitch et al., 2009; Song, Wang & Lv, 2016). The possible reason is that P. australis as the dominant species of coastal wetlands in China, having a strong ability in resisting environmental stress by dissipating excess excitation energy, which cannot be used in PSII photochemistry reaction as harmless heat through the xanthophyll cycle (Demmig-Adams et al., 1996; Lu et al., 2020; Zhang et al., 2015).

Figure 5 (A–D) Relationship between quantum yields of PSII photochemistry (ΦPSII) and quantum yields of regulated energy dissipation (ΦNPQ) in leaves of Phragmites australis under warming and precipitation change conditions.

Conclusions

In conclusion, the photosynthesis of P. australis during precipitation changing is dependent on non-stomatal limitation but not stomatal closure, which have a significant negative linear correlation with Chl a/b ratio and MDA content. At the same time, high temperature causes the biochemical limitation on photosynthesis, inhibits the positive effects of increased precipitation and aggravates the adverse effects of drought on photosynthesis of P. australis. Even though high temperature and drought (precipitation decrease) significantly decrease the carbon assimilation rate, P. australis still has a strong ability to protect itself from damages by transforming excess excitation energy into harmless heat. This study highlighted the significant role of precipitation change in regulating the photosynthetic performance of P. australis under elevated temperature conditions, which may help us to better understand the mechanisms of vegetation degradation and provide knowledge basis for the restoration of the vegetation in climate sensitive regions under the background of global change.

Supplemental Information

Supplemental Information 1 Raw data.

All measured parameters with three replicates.

Click here for additional data file.

We are grateful to Li Changjiang at Shandong Agricultural University for his assistance in the data collection and processing.

Additional Information and Declarations

Competing Interests

Author Contributions

Data Availability

The authors declare that they have no competing interests.

Linhong Teng conceived and designed the experiments, analyzed the data, prepared figures and/or tables, authored or reviewed drafts of the paper, and approved the final draft.

Hanyu Liu performed the experiments, prepared figures and/or tables, and approved the final draft.

Xiaonan Chu performed the experiments, authored or reviewed drafts of the paper, and approved the final draft.

Xiliang Song conceived and designed the experiments, performed the experiments, analyzed the data, prepared figures and/or tables, and approved the final draft.

Lianhui Shi conceived and designed the experiments, analyzed the data, authored or reviewed drafts of the paper, and approved the final draft.

The following information was supplied regarding data availability:

The raw measurements are available in the Supplemental File.

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
