# Peer review of "Effect of precipitation change on the photosynthetic performance of Phragmites australis under elevated temperature conditions"

_PeerJ, doi:10.7717/peerj.13087_

## Round 0.1 · original submission · Major Revisions

· Academic Editor

Major Revisions

Dear Dr. Song,

Please address the reviewers' comments

Kind Regards,
Valeria Spagnuolo

L. 127 The measurement of chlorophyll content was according to…the verb is missing. Replace to: The chlorophyll content was measured according to…

L. 136 The same, the verb is missing. Replace to The MDA content was measured according to…

Fig. 2 Usually, in the box-plots bars represent minimum and maximum values, the SD is at the bottom and top of each box and the horizontal line in each box is the mean or median value. If the bars represent the SD (as reported in the legend), what do the values of each box represent? Specify.

Reviewer 1 ·

Basic reporting

The research paper entitled “ Effect of precipitation change on the photosynthetic performance of Phragmites australis under elevated temperature conditions” has quantified an effect of precipitations on photosynthesis under elevated temperature.
The abstract of the paper is factual concrete, realistic, understandable, illuminates the main findings of the paper, and can serve as a stand-alone document that succinctly describes both procedures and conclusions.
The authors have put in effort to compile literature linking and highlighting the relation of photosynthesis under changing environmental and plant water regime. This part looks informative authors tried to write a good compilation of citations and review of the work, however, the majority of references are quite old and not very focused on mechanisms of photosynthesis. The philosophy of this work is good, applicable mostly in applied plant and agronomy sciences. The understanding of mechanisms is limited.
The paper might be a hot topic for the next experimental research in model crop plants, therefore, contributing to its merit for acceptance in PeerJ.

Experimental design

The experimental methodology was appropriate, enough strong and scientific, and the analyses were done correctly. The authors used very progressive techniques and protocols and described used protocols perfectly.

Validity of the findings

The novelty of this manuscript is not very high, however, the paper brings new aspects and news. I would like to encourage the authors to discuss more ecophysiological aspects mechanisms, including impact on plant photosynthesis and primary processes of biomass production. The authors should know about classical papers (Cornic G., Lawlor D., Chaves M., Brestic M., Loretto G., Zivcak M. etc.) about the flexibility of photosynthesis apparatus under drought stress and the stomatic and non-stomatic inhibition of photosynthesis, which is a key limitation until 25-30% of leaf relative water content.
The paper adheres to appropriate reporting guidelines and community standards for data availability and is presented in an intelligible fashion and is written in standard English.

Additional comments

Several sections in the manuscript lack experimental evidence as well important references of authoritative articles. Authors were unable to generate their strong own synthesis of information in a scientifically correct manner. The authors could explain more about the mechanisms of plant sensitivity to water deficit. Authors could involve also new aspects of plant responses to water deficit and also modern phenotyping approaches used in precise agriculture, including new references.
In the current version, the review could be improved toward being more balanced, structured and critical. The authors have to discuss more in functioning photosynthetic apparatus in a changing environment.
The biggest gap in this paper is a lower understanding of the research of photosynthesis under stress.
The authors should improve the interpretations and discuss their results more properly.
Paper bring many new aspects and the novelty of the paper is OK, but I would like to invite authors to discuss more eco-physiological aspects and molecular mechanisms using new references:
• Drought and Heat Stress in Cool-Season Food Legumes in Sub-Tropical Regions: Consequences, Adaptation, and Mitigation Strategies. Plants 2021, 10, 1038. https://doi.org/10.3390/plants10061038
• Kernel Water Relations and Kernel Filling Traits in Maize (Zea mays L.) Are Influenced by Water-Deficit Condition in a Tropical Environment. Front. Plant Sci. 12:717178. doi: https://dx.doi.org/10.3389/fpls.2021.717178
• Consequences and Mitigation Strategies of Abiotic Stresses in Wheat (Triticum aestivum L.) under the Changing Climate. Agronomy 2021, 11, 241. https://doi.org/10.3390/agronomy11020241
• Photosynthesis research under climate change. Photosynth Res (2021). https://doi.org/10.1007/s11120-021-00861-z

Reviewer 2 ·

Basic reporting

Please see the comments below.

Experimental design

Please see the comments below.

Validity of the findings

Please see the comments below.

Additional comments

Manuscript 67054v1

The manuscript reported the results concerning the changes in the photosynthetic traits in plants of Phragmites australis to water status change and high temperature. The main results are 1) increased precipitation can promote the photosynthetic capacity, whereas water deficit largely decreases it. 2) elevated temperature weakens positive effects of increased precipitation, and enhances the adverse effects of drought. 3) non-stomatal limitation can mainly contribute to carbon assimilation rate. 4) the xanthophyll cycle-mediated thermal dissipation may play a major role in PSII photoprotection, i.e., an adaptive traits. This study seems to be designed well, and the methods and data are appropriate. This main finding can be useful for the better understanding the plant stress physiology under the climatic change. Thus, it can be published. However, several issues should be addressed or revised as follows:


Abstract
‘To better understand how precipitation change and climate warming impact the photosynthetic performance of Phragmites australis (P. australis)’ is not the method topic, but object topic. It can be moved in the object part above. Or it could be changed accordingly.
“The combination of high temperature and precipitation change had more detrimental effect on P. australis than a single factor” seems to not be clear. What is precipitation change? it may be a deceased precipitation (drought)? Please change it.
In the section, the conclusion and/or application expressions seems to be lacked. Please add it.
Introduction
Line 2: “Global warming caused by high levels of greenhouse gas emission” can be change to “Global warming mainly caused by high levels of greenhouse gas emission”
Line 5: “IPCC, 2014” can be updated to the one “IPCC, 2021/2021?”. Please search, check and confirm it.
Line 21: “It’s widely accepted that”? “It’s” should not be used in its informal case, and may be changed to ‘it is’. Please check this case throughout the ms test.
Line 63-64: “Their spatial distribution is mainly limited by heat and water” seems to be not logical, and can be changed “Their spatial distribution is mainly limited by heat shortage and water deficit”?
Materials & Methods
Lines 78-79: How many year did the data obtain the annual average temperature and precipitation? 30 years? Please add this information.
Lines 87-96: some information could be added in details: such as light condition, the environmental chamber size.
Results
This section seems to be overstated a little bit, and it can be shortened to be concise. For instance, for photosynthetic parameters, lines 172-196, this part seems to be tedious, it can be shortened more to be readable more.

Conclusions
This part have main results, conclusion, and application expressions. However, the latter seems to be lacked. Please add this information.

---

## Round 0.2 · Minor Revisions

· Academic Editor

Minor Revisions

The authors fulfilled all requests of the reviewers; however, I still noticed some minor errors or inaccuracies, which can be found on the attached annotated pdf.

Reviewer 2 ·

Basic reporting

Yes.

Experimental design

Yes.

Validity of the findings

Yes.

Additional comments

The newly version has been prepared well, and the responses to the comments is ample. Thus, I can recommend it for publication after carefully checking it throughout.

---

## Round 0.3 · accepted · Accept

· Academic Editor

Accept

The authors have fulfilled all the requests raised by the Reviewers and the Editor.

At line 115 of the tracked_manuscript is reported "This main aims of the work were to investigate...". I suggest changing this to "The main aims of the work were....".